A synthesis tree of the Copepoda: integrating phylogenetic and taxonomic data reveals multiple origins of parasitism

Bernot James P. bernotj@si.edu 1 2
Boxshall Geoffrey A. 3
Crandall Keith A. 1 2
1 Department of Invertebrate Zoology, Smithsonian National Museum of Natural History , Washington , DC , United States of America
2 Computational Biology Institute, Milken Institute School of Public Health, George Washington University , Washington , DC , United States of America
3 Department of Life Sciences, Natural History Museum , London , United Kingdom
Waiho Khor
Electronic publication date: 2021 Aug 18
Publication date: 2021
Volume: 9
Electronic Location ID: e12034
Received 2021 Jun 3; Accepted 2021 Aug 1
Copyright: ©2021 Bernot et al.
Copyright year: 2021
Copyright holder: Bernot et al.
License: This is an open access article distributed under the terms of the Creative Commons Attribution License, which permits unrestricted use, distribution, reproduction and adaptation in any medium and for any purpose provided that it is properly attributed. For attribution, the original author(s), title, publication source (PeerJ) and either DOI or URL of the article must be cited.
License URL: https://creativecommons.org/licenses/by/4.0/

Keywords: Copepoda, Copepod, Parasite, Evolution, Phylogeny, Parasitism, Synthesis, Taxonomy

Funding: NSF PRFB DBI 2010898 Society of Systematic Biologists mini-ARTS Award GWU Knowledge in Action Career Internship Fund Smithsonian NMNH Reed Fellowship for Copepod Research This material is based upon work supported by the NSF Postdoctoral Research Fellowships in Biology Program under Grant No. 2010898 to JPB. This work was also supported in part by funding from the Society of Systematic Biologists mini-ARTS award, the GWU Knowledge in Action Career Internship Fund, and the Smithsonian NMNH Reed Fellowship for Copepod Research to JPB. There was no additional external funding for this study. The funders had no role in study design, data collection and analysis, decision to publish, or preparation of the manuscript.

==============================
The Copepoda is a clade of pancrustaceans containing 14,485 species that are extremely varied in their morphology and lifestyle. Not only do copepods dominate marine plankton and sediment communities and make up a sizeable component of the freshwater plankton, but over 6,000 species are symbiotically associated with every major phylum of marine metazoans, mostly as parasites. Unfortunately, our understanding of copepod evolutionary relationships is relatively limited in part because of their extremely divergent morphology, sparse taxon sampling in molecular phylogenetic analyses, a reliance on only a handful of molecular markers, and little taxonomic overlap between phylogenetic studies. Here, a synthesis tree method is used to integrate published phylogenies into a more comprehensive tree of copepods by leveraging phylogenetic and taxonomic data. A literature review in this study finds fewer than 500 species of copepods have been sampled in molecular phylogenetic studies. Using the Open Tree of Life platform, those taxa that have been sampled in previous phylogenetic studies are grafted together and combined with the underlying copepod taxonomic hierarchy from the Open Tree of Life Taxonomy to make a synthesis phylogeny of all copepod species. Taxon sampling with respect to molecular phylogenetic analyses is reviewed for all orders of copepods and shows only 3% of copepod species have been sampled in phylogenetic studies. The resulting synthesis phylogeny reveals copepods have transitioned to a parasitic lifestyle on at least 14 occasions. We examine the underlying phylogenetic, taxonomic, and natural history data supporting these transitions to parasitism; review the species diversity of each parasitic clade; and identify key areas for further phylogenetic investigation.

Introduction

The Copepoda is a diverse, monophyletic group of crustaceans comprising 14,485 valid species (Walter & Boxshall, 2021) whose phylogenetic relationships remain poorly resolved for a number of reasons. First, copepods have extremely diverse body plans (e.g., size range 0.5 mm–300 mm; 0 to 11 pairs of appendages; segmented or lacking external segmentation) and assessing homology across these disparate forms is challenging (e.g., Fig. 1) (Huys & Boxshall, 1991; Boxshall & Halsey, 2004). This is particularly true for the over 5,000 species of parasitic copepods, many of which have reduced or secondarily lost structures traditionally used in copepod classification (i.e., setal elements, segmentation, and entire appendages) (Kabata, 1979; Huys & Boxshall, 1991; Boxshall & Halsey, 2004).

Figure 1 Photographs showing diversity of parasitic copepod body plans.

(A) Caligidae. (B) Dichelesthiidae. (C) Pennellidae. (D) Lernaeopodidae. (E) Philichthyidae. Photos courtesy of the Natural History Museum, London. Image credit: K. Norris, Natural History Museum, London.

Second, challenges in molecular phylogenetics have also limited our understanding of copepod phylogeny. Many copepods were historically preserved in formalin, which has limited the availability of specimens for molecular analysis. There is also a limited assortment of molecular markers available for copepods. The majority of molecular phylogenetic studies of copepods are single gene studies using 18S or CO1, and in the best cases, two to four other markers (Table 1). Given that copepods are estimated to have diverged 375–450 million years ago (mya) (Schwentner et al., 2017), it is challenging to reconstruct both deep and shallow levels of the phylogeny with a small number of relatively short markers. Moreover, the most widely used marker in copepod phylogenetics is 18S and it contains large indels in a number of taxa which can make alignment and phylogenetic reconstruction more challenging (Huys et al., 2012; Cornils & Blanco-Bercial, 2013; Wu, Xiong & Yu, 2016). In addition, few copepod species have sequence data available in NCBI (approximately 1,400 of 14,485 species) and many of the NCBI sequences are not identified to species level (e.g., Copepod sp. 142, Cyclopoida gen. sp., etc.) (Clark et al., 2016; GenBank, 2021). Furthermore, copepod systematists have typically specialized on one of five ecological groups: marine planktonic copepods, copepods inhabiting marine sediments, freshwater copepods, parasitic copepods associated with fish hosts, or parasitic copepods associated with invertebrate hosts (Huys & Boxshall, 1991; Boxshall & Halsey, 2004), and while these groups are intermingled, the focus on particular ecological groups has limited the integration of copepod phylogenies into a cohesive tree of life.

Table 1 Studies used for copepod synthetic tree.

Rank	Study	Figure	Focal taxon	Markers	# IG taxa	Tree method	
1	Khodami et al. (2019)	Figure 2	Cyclopoida	18S, 28S, CO1	121	BI, ML	
2	Bradford-Grieve et al. (2017)	Figure 113	Megacalanidae	18S, 28S, 5.8S, CO1, H3, ITS1, ITS2	11	BI, ML, MP, NJ	
3	Cornis & Blanco Bercial(2013)	Figure 3	Paracalanidae	18S, CO1, H3	22	BI, ML	
4	Bradford-Grieve, Boxshall & Blanco-Bercial (2014)	Figure 12	Calanoida	18S, 28S, CO1, CytB	38	BI, ML	
5	Blanco-Bercial, Bradford-Grieve & Bucklin (2011)	Figure 2	Calanoida	18S, 28S, CO1, CytB	32	BI, ML	
6	Wyngaard, Hołyńska & Schulte (2010)	Figure 2	Mesocyclops	18S, ITS2	10	ML	
7	Huys et al. (2012)	Figure 2	Cyclopoida	18S	41	BI, MP	
8	Huys et al. (2007)	Figure 1	Copepoda	18S	49	BI, MP	
9	Huys, Mackenzie-Dodds & Llewellyn-Hughes (2009)	Figure 1	Harpacticoida	18S	31	BI, MP	
10	Lozano Fernandez et al. (2019)	Figure 2	Pancrustacea	244 orthologs	9	BI	
11	Oakley et al. (2013)	Figure 12	Pancrustacea	transcriptomes	7	BI, ML	
12	Eyun (2017)	Figure 3	Copepoda	24 nuclear genes	9	BI, ML	
13	Huys et al. (2006)	Figure 1	Copepoda	18S	47	BI, MP	
14	Krajíček et al. (2016)	Figure 2	Cyclops	12S, 16S, 18S, CO1, CytB, ITS1	15	ML	
15	Minxiao et al. (2011)	Figure 9	Copepoda	mt genome	6	BI	
16	Laakmann, Auel & Kochzius (2012)	Figure 2a	Aetideidae, Euchaetidae	18S, 28S, CO1, ITS2	14	ML	
17	Zagoskin et al. (2014)	Figure 2	Cyclopoida	28S	16	ML, ME	
18	Machida et al. (2006)	Figure 3	Neocalanus	18S, 28S, CO1, ND4, ND6	7	BI, ML, MP, NJ	
19	Hirai, Shimode & Tsuda (2013)	Figure 3	Calanoida	28S	58	BI, ML	
20	Takenaka et al. (2012)	Figure 3	Copepoda	18S	35	ML	
21	Adamowicz et al. (2010)	Figure 3	Centropagidae	16S	25	BI	
22	Adamowicz et al. (2007)	Figure 3	Centropagidae	CO1	14	BI	
23	Bucklin et al. (2003)	Figure 3	Pseudocalanus	18S, CO1	7	NJ	
24	Bucklin et al. (2003)	Figure 2	Neocalanus	18S, CO1	7	NJ	
25	Bucklin et al. (2003)	Figure 1	Clausocalanus	18S, CO1	14	NJ	
26	Bucklin et al. (2003)	Figure 4	Calanidae, Clausocalanidae	18S, CO1	12	NJ	
27	Taniguchi et al. (2004)	Figure 3	Neocalanus	16S, 18S	11	NJ	
28	Taniguchi et al. (2004)	Figure 2	Calanoida	16S, 18S	19	NJ	
29	Braga et al. (1999)	Figure 4	Calanoida	28S	11	MP	
30	Braga et al. (1999)	Figure 6	Euchitidae	16S	8	MP	
31	Mayor et al. (2010)	Figure 3	Cyclopoida	CO1	19	NJ	

Given that copepod systematics has generally been approached from a focused ecological perspective, an overall view of copepod phylogeny requires an integration of these independent efforts. Three main classes of methods are available for integrating phylogenies into a more comprehensive tree, i.e., supertree, supermatrix, or synthesis tree (Ewers-Saucedo et al., 2019). Supertree methods code phylogenies into a new matrix to be analyzed by phylogenetic methods, but this requires a large number of the same taxa to be present in each study in order to effectively integrate multiple phylogenies into a single tree (Bininda-Emonds, 2004). This requirement would greatly reduce the number of studies incorporated, given the relatively low taxonomic overlap between published copepod molecular studies. Supermatrix approaches require the same characters (i.e., morphology and/or molecular loci) to be shared across studies (McMahon & Sanderson, 2006), a requirement that is also problematic for available copepod morphological and molecular data. Given this lack of matching data (i.e., matching taxa in supertree methods and matching characters in supermatrix methods), the synthesis tree method is employed here.

A synthesis tree grafts together published phylogenies to produce a more comprehensive phylogeny including all taxa that have been sampled (Hinchliff et al., 2015; Owen et al., 2015; Redelings & Holder, 2017; Rees & Cranston, 2017; Ewers-Saucedo et al., 2019). This method can also include taxa that have not been previously sampled in other phylogenetic analyses by incorporating taxonomic information using the Open Tree of Life Taxonomy (OTT) (Hinchliff et al., 2015; Redelings & Holder, 2017; Rees & Cranston, 2017). This works by taking advantage of the tree-like properties of the Linnaean classification system; each taxonomic category is a polytomy within each higher taxonomic category such that all species are a polytomy within a genus, all genera are a polytomy within a family, and so on throughout the taxonomic hierarchy. Given that taxonomists have strived to establish monophyletic taxa, the taxonomic hierarchy is a useful null hypothesis to incorporate taxa that have never been sampled in a phylogenetic analysis. This approach has the additional benefits of highlighting taxa that have limited phylogenetic information, being computationally fast, and easily revised with the incorporation of new studies (Redelings & Holder, 2017).

Here we construct a synthetic phylogeny of the Copepoda using published molecular phylogenies and the OTT to build a synthesis phylogeny of all known copepod species. The goals of this study are to integrate published copepod molecular phylogenies in order to synthesize current information on copepod phylogeny, to identify parts of the copepod tree of life that require additional sampling effort, and to use the resulting phylogeny to estimate the number of times copepods have evolved a parasitic lifestyle. Improving our understanding of copepod evolution is of key interest because copepods are an excellent system for studying the evolution of parasitism: they have evolved to be parasitic multiple times, have an exceptionally broad host range spanning 14 phyla of metazoans, and exhibit extreme morphological diversity (e.g., Fig. 1) (Boxshall & Halsey, 2004).

Materials and Methods

A literature review for molecular phylogenies of copepods was conducted using Google Scholar and the follow search term <“copepod AROUND(5) phylogeny” molecular>. As of June 01, 2021 this produced 178 search results, which were screened individually for copepod phylogenies. Despite containing the search terms, the majority of search results did not contain original molecular phylogenies of copepods. Other relevant literature was added manually, such as phylogenetic studies at higher taxonomic levels above copepods that were lacking the search terms. Of the approximately 75 results that did contain original phylogenies, the following exclusion criteria were applied. If a study contained only a subset of taxa and loci that were sampled in another study using comparable methods, the smaller phylogeny was excluded. Studies that contained only intraspecific taxon sampling or that had low support (<70% BS or <0.95 PP) at all interspecific nodes were excluded. Those with highly disjunct taxon sampling, such as unrelated species from a particular region or habitat, were also excluded to avoid sampling bias that, in the absence of more even taxon sampling, would lead to the accumulation of these taxa towards the base of the synthesis tree. Phylogenies from studies that contained multiple single gene phylogenies with high conflict, no combined analyses, and no objectively preferred topology were also excluded to avoid subjectivity. Finally, unrooted phylogenies were excluded because the synthesis tree method requires rooted trees.

All phylogenies were reconstructed manually in Mesquite v3.31 (Maddison & Maddison, 2017) based on the published figures and imported into the Open Tree of Life. The phylogenies were grafted together individually and with underlying taxonomic information (OTT v3.2) (Rees & Cranston, 2017) using the propinquity pipeline (Redelings & Holder, 2017). As required by that pipeline, the studies were ranked; ranking was based on (1) density of taxon sampling of the focal clade (sparser taxon sampling and geographic vs phylogenetic scope ranked lower), (2) number of loci, and (3) methodology (neighbor joining and parsimony methods ranked lower than ML and BI methods). The specific phylogeny figures used from each study and their relative rankings are shown in Table 1. Two phylogenies were produced using the propinquity pipeline: a grafted tree of all input taxa (grafted_solutions_ottnames.tre, Data S1) and a grafted tree including those taxa as well as taxonomic data for all copepod species (labelled_supertree_simplified_ottnames.tre) (Data S2). Ancestral state reconstruction analyses were conducted on the grafted tree of all input taxa from the propinquity analysis (i.e., grafted_solutions_ottnames.tre) by scoring each taxon as free-living or parasitic in PAUP* (v4.0a168) (Swofford, 2003) under a parsimony reconstruction method using ACCTRAN and DELTRAN. Parsimony was the only possible ancestral state reconstruction method because trees produced by the propinquity pipeline do not contain branch lengths. An additional parsimony ancestral state reconstruction was also carried out in PAUP* under an irreversible parsimony model that did not allow for reversals from parasitic to free-living (DEFTYPE=IRREV) since the evolution of parasitism has been speculated to be an irreversible transition (e.g., Futuyma & Moreno, 1988; Cruickshank & Paterson, 2006; Goldberg & Igić, 2008. The resulting phylogenies (Figs. 2 and 3) were visualized using the Interactive Tree of Life v3 (Letunic & Bork, 2016).

Figure 2 Copepod synthesis grafted tree without taxonomic data.

Phylogeny of 365 copepod taxa from 31 published phylogenies. Copepod orders are colored. Parasitic taxa are shown with red branches. Dotted red branches indicate taxa that contain free-living and parasitic species, but for which parasitic species have not yet been sampled. Bold black branches in the poecilostome clade indicate two reversals from free-living to parasitic identified in ancestral state reconstruction analyses. Red arrowheads indicate additional transitions to parasitism in constrained ancestral state reconstruction analysis not allowing reversals from parasitic to free-living.

Figure 3 Synthesis tree of all copepod species.

Phylogeny of all copepod species including 31 published phylogenies and grafted taxonomic hierarchy. Copepod orders are colored and parasitic taxa (n = 14) are labeled and shown with red branches. Due to the size of the tree, clades with 200–500 tips are collapsed.

In order to categorize parasitic copepods among the many copepods associated with other organisms, an operational definition of parasitism was applied to demarcate parasites along the continuum of symbioses (i.e., mutualism to commensalism to parasitism). Here, we follow Poulin (2007) and define parasitism as an interspecific symbiotic association between two organisms in which one organism (the parasite) has a nutritional dependence on the other (the host) for a prolonged period of time and has a negative impact on the fitness of the host, differing from Poulin (2007) only in the qualification of for a prolonged period of time to ensure the association is durable and to exclude micropredators.

Results

Copepod phylogeny and synthesis tree

In total, 26 relevant publications encompassing 31 phylogenies based on molecular data were identified and added to the Open Tree of Life online curatorial system where they are publicly available (https://tree.opentreeoflife.org/curator/profile/jbernot/copepoda) (McTavish et al., 2015). The synthesis tree of copepods includes 15,763 terminals (Fig. 3), substantially more than the 14,485 curated copepod species documented in the World of Copepods database (Walter & Boxshall, 2021). The approximately 1,200 additional terminals come primarily from NCBI registrants that include putatively identified species (e.g., those with a “cf.” designation), undescribed species (e.g., Caligus sp. 1), and environmental samples not identified to the species level (e.g., Cyclops sp. environmental sample).

Molecular phylogenetic information was available for 365 copepod taxa (Table 2). In total, only 3% (365/14,485) of copepod species have been examined in a molecular phylogenetic context. The phylogeny resulting from grafting together the published phylogenies is shown in Fig. 2. Some copepod taxa have been sampled in greater detail, while others have been largely unexplored in terms of their phylogenetic relationships. The most well-sampled order of copepods is the Calanoida. While only 6% (173/2,709) of calanoid species have been included in molecular phylogenetic analyses, the limited species-level taxon sampling has spanned 70% (31/44) of calanoid families. The three orders Siphonostomatoida, Harpacticoida, and Cyclopoida encompass 80% (11,449/14,485) of copepod species, but few species in each of these orders have ever been studied in a molecular phylogenetic context. In the Cyclopoida, only 3% (141/4,500) of species spanning 39% (40/103) of families have been included in a molecular phylogenetic analysis. The Harpacticoida and Siphonostomatoida are the most poorly sampled with <1% (29/4,687) of harpacticoid species and 17% (14/79) of families sampled and 1% (23/2,262) of siphonostomatoid species and 26% (15/58) of families of sampled.

Table 2 Phylogenetic sampling of the copepod orders.

Order	Number of species	Number of taxa in synthesis tree	%	
Calanoida	2,709	169	6.2%	
Cyclopoida	4,500	141	3.2%	
Gelyelloida	2	0	0.0%	
Harpacticoida	4,771	30	0.6%	
Misophrioida	37	0	0.0%	
Monstrilloida	173	3	1.7%	
Mormonilloida	4	0	0.0%	
Platycopioida	11	0	0.0%	
Siphonostomatoida	2,262	23	1.0%	
Total Copepoda	14,485a	365	2.5%	
Notes.

a Including 16 species of uncertain ordinal status.

Transitions to parasitism

Despite the limited taxon sampling of copepods in molecular phylogenetic studies, we identify multiple clades of parasitic copepods. Evidence supporting each parasitic clade is presented in one of three ways: direct phylogenetic support, indirect phylogenetic support, and taxonomic support based on morphology. Direct phylogenetic support consists of cases in which the parasitic taxon has been sequenced and molecular phylogenetic analysis shows it is distinct from all other parasitic clades in the ancestral state reconstruction analysis (solid red branches in Fig. 2). We refer to other clades as having indirect phylogenetic support (dotted red branches in Fig. 2). In these cases, a taxon that contains both free-living and parasitic species was included in a molecular phylogenetic analysis, but the parasitic species themselves have not been sampled. Finally, we identify a number of transitions to parasitism in taxa that have not been sampled in molecular phylogenetic analyses, but that have taxonomic support based on morphology suggesting their independence from all other parasitic clades.

There is direct phylogenetic support for seven transitions to parasitism comprising the following clades: Alteuthellopsis Lang, 1948, Ameiridae Boeck, 1865; Lernaeidae Cobbold, 1879; Notodelphyidae Dana, 1853 (plus related families of tunicate parasites); the poecilostome cyclopoids previously classified as the “Poecilostomatoida”; a single origin in Siphonostomatoida + Monstrilloida; and Thaumatopsyllidae Sars, 1913 (solid red branches in Fig. 2). The unconstrained parsimony analyses supported two reversals from parasitic to free-living lifestyles (bold black branches in Fig. 2). ACCTRAN and DELTRAN parsimony analyses were identical. An analysis constrained to not allow reversals from parasitism back to free-living supported eight, rather than 1, transitions to parasitism in the poecilostome cyclopoids (red arrowheads in Fig. 2). Many parasitic copepods remain to be sampled in phylogenetic analyses.

There were also some indirect phylogenetic data supporting four additional parasitic clades (dotted red branches in Fig. 2). These three instances comprise the harpacticoid taxa: Aglaogastes cnidicus Humes, 1981; Huys, 2016; Canuellidae Lang, 1940, Thalestridae Claus, 1862, and the cyclopoid genus Eucyclops Claus, 1893, all of which represent independent clades in this analysis, but for which only free-living species were sampled in published molecular phylogenetic analyses. Assuming the familial and generic placement of the parasitic members of these taxa is correct, the dotted lines in Fig. 2 indicate four additional transitions to parasitism in the copepod phylogeny.

Taxonomic data for all copepod species based on morphology adds an additional data layer to this study and provides support for three more transitions to parasitism. The harpacticoid taxa Balaenophilus Aurivillius, 1879, Cholidyinae Boxshall, 1979, and Laophontidae Scott, 1904 each contain parasitic species, but none of these taxa have been sampled in a molecular phylogenetic analysis. If the current taxonomic hierarchy is correct such that these three taxa are indeed distinct from the 11 parasitic clades mentioned above, there are at least 14 independent clades of parasites within the Copepoda (Fig. 3). By including taxonomic information for all copepod species, we show the diversity of the parasitic clades varies from a single species in three cases to two instances of over 2,000 parasitic species in the large siphonostomatoid and poecilostome clades (Table 3, Fig. 3).

Table 3 Diversity of parasitic copepod clades by order.

Taxon	Number of parasitic species	
Siphonostomatoida	2,262	
Monstrilloida	173	
Cyclopoida		
poecilostomes	2,235	
Notodelphyidae + Ascidicolidae + others	699	
Lernaeidae + Ozmanidae	129	
Thaumotopsyllidae	5	
Eucylops bathanalicola	1	
Harpacticoida		
Ameiridae	4+	
Balaenophilus	2–3	
Canuellidae	1+	
Cholidyinae	14	
Laophontidae	4+	
Peltidiidae	2+	
Tegastidae	1+	
Thalestridae	4	

Discussion

Copepod phylogeny and synthesis tree

The phylogenetic sampling of copepods has not been distributed evenly across the copepod tree. The copepod orders with the greatest diversity, in terms of species richness, morphological variation, and lifestyles, remain the most poorly sampled, that is, the Cyclopoida, Harpacticoida, and Siphonostomatoida. These three orders encompass nearly all parasitic copepod species except for the 173 parasitic species in the Monstrilloida, which is thought to be closely allied with, or even nested within, the Siphonostomatoida (see Huys et al., 2007).

To better understand copepod evolution, it is necessary to sample many higher taxa (e.g., copepod orders and families) that have not yet been included in molecular phylogenetic analyses. This is particularly true for the four orders Gelyelloida, Misophrioida, Mormonilloida, and Platycopioida that have never been sampled in molecular phylogenetic studies relative to the other copepod orders. In addition, the Cyclopoida, Harpaticoida, and Siphonostomatoida, which comprise >80% of copepod diversity, require greater sampling effort; only 40/103, 14/79, and 15/58 families, respectively, have been included in phylogenetic studies. Furthermore, phylogenetic relationships of the Cyclopoida and Harpacticoida are key to our understanding of the evolution of parasitism, and, particularly in the cyclopoids, morphological evolution and the colonization of freshwater.

A notable artefact of the synthesis method here is the placement of Cyclopicina in a basal polytomy comprising the Siphonostomatoida, Harpacticoida, and Cyclopoida (Fig. 2). Khodami et al. (2019) showed this genus to be sister to all other cyclopoids they sampled, but because this genus was also sampled in the phylogeny of Huys et al. (2006) that sampled only cyclopoids and siphonostomatoids, the synthesis method used here placed the Cyclopicina in a more basal position due to the differing positions of this taxon in these two studies. This apparent conflict is due primarily to nonoverlapping taxa in these studies rather than conflicting phylogenetic signal. This is an artefact of the synthesis method; in reality, the placement of the Cyclopicina should be at the base of the Cyclopoida. Greater overlap in taxon sampling across phylogenetic studies will reduce this issue in future synthesis phylogenies.

A number of taxonomic issues remain to be addressed in the Copepoda. Comparing phylogenetic and taxonomic hypotheses here shows that four orders and many families have uncertain placement or are polyphyletic. Both of these issues lead to the accumulation of these taxa towards the base of the copepod synthesis tree. This is because taxa that are included in a polyphyletic taxon that have not themselves been sampled in a phylogenetic study (i.e., that do not have corresponding phylogenetic information) can only be placed at the next higher taxonomic rank in the current synthesis tree methodology. This is evident in the Harpacticoida (Fig. 3) where a large basal polytomy corresponds to the majority of harpacticoid families, which have yet to be sampled in a molecular phylogenetic study. Likewise, since the relationships of the Gelyelloida, Misophrioida, and Mormonilloida relative to the other copepod orders have not been examined in molecular phylogenetic studies, these three orders are part of a basal polytomy in the superorder Podoplea. A number of cyclopoid families of uncertain placement, as well as genera from podoplean families that were found to be paraphyletic also accumulated at the base of the Podoplea (gray branches in Fig. 3). These are artefacts of the synthesis tree method. While Khodami et al. (2019) established suborders for most cyclopoid taxa, this revised taxonomy, already adopted by WoRMS (Walter & Boxshall, 2021), has not yet been incorporated into the Open Tree of Life Taxonomy (OTT). Once this taxonomic revision is incorporated into the OTT, the orphaned cyclopoid families will form a polytomy at the base of the Cyclopoida rather than the Podoplea. The incorporation of additional phylogenetic data into the copepod synthesis tree for taxa not previously sampled (e.g., many of the cyclopoid, harpacticoid, and siphonostomatoid families) will further resolve basal polytomies in the synthesis phylogeny and lead to a more bifurcated synthesis tree.

Transitions to parasitism

Despite the taxon sampling limitations of copepods in molecular phylogenetic analyses, the synthetic phylogeny here provides insights into copepod phylogeny and evolution, particularly the multiple origins of a parasitic lifestyle among copepods. Mapping parasitic taxa documented in the literature onto the synthetic phylogeny reveals parasitism has evolved in the Copepoda at least 14 times (Fig. 3). There are direct phylogenetic data supporting at least seven independent transitions to parasitism (solid red branches in Fig. 2). Current data suggest a single transition to parasitism appears to have given rise to the wholly parasitic order Siphonostomatoida with the parasitic order Monstrilloida nested within it. Members of these orders are obviously parasitic with negative effects on their hosts, which range from sponges to fish, and include a number of economically important parasites in aquaculture (Johnson et al., 2004).

Within the Cyclopoida there is direct phylogenetic support for four clades of parasites and indirect support for a fifth clade. The clades with direct support are as follows. The largest is a clade of more than 2,000 species of poecilostome copepods (Walter & Boxshall, 2021), nested within the suborder Ergasilida, whose hosts span nearly all major groups of marine metazoans including fish. All but a few species are obviously parasitic, and those that are not are often associated with gelatinous zooplankton (Heron, 1973; Gasca, Suárez-Morales & Haddock, 2007). Next is the Notodelphyidae, comprising a group of closely related families of parasites (some are arguably commensals) of solitary and colonial ascidians (Walter & Boxshall, 2021). Morphological phylogenetic analysis by Ho (1994) separated the ascidicolid taxa from the Notodelphyidae + Archinotodelphyidae but we have conservatively counted this as a single transition encompassing the tunicate parasitizing copepods because the ascidicolid taxa have not been examined in molecular phylogenetic analyses. The third clade is the Lernaeidae, an exclusively parasitic family of mostly mesoparasites on freshwater fishes (Walter & Boxshall, 2021). The Ozmanidae, which is associated with freshwater snails, likely belongs to this same clade (Ho, 1994). The fourth clade with direct support is the Thaumatopsyllidae, a unique family composed of five species that, as nauplii, are endoparasites in the stomach of brittle stars (Boxshall & Halsey, 2004; Khodami et al., 2019; Walter & Boxshall, 2021). The only cyclopoid clade with indirect phylogenetic support is Eucyclops bathanalicola, the lone parasitic member of Eucyclops. Evidence of its parasitic lifestyle comes from its location in the mantle cavity of its snail host and its highly modified morphology: it attaches with modified, clawed maxillules and maxillae, and its reduced maxillipeds, which are used for feeding in free-living Eucylops but are vestigial lobes in this species (Boxshall & Strong, 2006).

Interestingly, two small clades of free-living copepods (the Oncaeidae and a second clade consisting of Sapphirinidae + Corycaeidae + Pachos) are nested within the large poecilostome clade in the phylogeny of Khodami et al. (2019) and thus the synthesis phylogeny here. Strictly interpreting this topology, either the single transition to parasitism for the entire clade of poecilostome cyclopoids includes two reversals to free-living lifestyles (bold black branches in Fig. 2), or there are eight independent transitions to parasitism (red arrowheads in Fig. 2). Both ACCTRAN and DELTRAN parsimony-based ancestral state reconstructions predicted reversals from parasitic to free-living in this phylogeny. While the evolution of parasitism has been considered an irreversible transition (e.g., Futuyma & Moreno, 1988; Goldberg & Igić, 2008, recent studies have called this derivative of Dollo’s law into question (Cruickshank & Paterson, 2006; Klimov & OConnor, 2013). The reversals from parasitism to free-living in both of these instances in the Cyclopoida are similar to that observed in psoroptidian mites by Klimov & OConnor (2013). Here, the free-living taxa (i.e., Corycaeidae, Oncaeidae, Pachos, and Sapphirinidae), though planktonic, are often associated with gelatinous zooplankton (Heron & Damkaer, 1978; Gasca, Suárez-Morales & Haddock, 2007) and have mouthparts adapted for surface feeding rather than particle feeding (Heron, 1973; Boxshall & Halsey, 2004). Alternatively, the possibility of eight, rather than one, independent evolutions of parasitism in poecilostome copepods is not unprecedented given the 11 other instances across the Copepoda. Denser taxon sampling is needed to more definitively evaluate transitions to parasitism in poecilostome cyclopoids since only 23/67 poecilostome families (75/2,425 species) have been sampled in molecular phylogenetic analyses.

Within the Harpacticoida there is direct phylogenetic support for two independent clades of parasites, indirect phylogenetic support for three additional clades, and taxonomic support for three more parasite clades for a total of eight parasitic harpacticoid lineages. In most of these cases, the nature of the association of harpacticoids with their hosts is less obviously parasitic relative to other copepods. Direct phylogenetic data support two clades of parasites, one in the Ameiridae and the other in the Peltidiidae. There are a number of potentially parasitic genera in the Ameiridae including Abscondicola Fiers, 1980, Antillesia Humes, 1958, Nitokra Boeck, 1865 and possibly a few other genera such as Cancrincola Wilson, 1913. Since the phylogeny of the Ameiridae is not well known, we conservatively counted the Ameiridae as a single transition represented here by Nitokra (Fig. 2). We find convincing evidence of parasitism in N. bdellurae (Liddell, 1912) and N. spinipes Boeck, 1875; other species of Nitokra associated with hosts including crayfish and marine isopods may be parasitic as well. Nitokra bdellurae is an egg parasite or parasitoid of the turbellid flatworms and N. spinipes is parasitic on medusae of Aurelia where it is found in huge numbers—up to 1,030 from a single medusae by Humes (1953)—in pits thought to be excavated by the copepods (Liddell, 1912; Humes, 1953, 1981). Available natural history data suggest the monotypic genera Abscondicola and Antillesia are additional ameirid parasites. Both utilize inland gecarcinid land crabs and given that 14 to >250 individuals were found at 100% prevalence by Fiers (1990), it is hard to believe these morphologically modified copepods do not have a negative effect on host fitness given the intensity of infection. Second, there is direct phylogenetic support for parasitism in a number of copepods from the Peltidiidae that are associated with invertebrates. While the exact nature of their relationship with hosts is unclear in most cases, we found compelling evidence of parasitism based on intense infections of corals with Alteuthellopsis corallina Humes, 1981;(Humes, 1981) collected hundreds of these copepods per coral head. Humes (1985) suggested coral-associated copepods fed on coral mucus, which would make them parasites under our definition assuming mucus feeding has a negative effect on corals.

Indirect phylogenetic data support three additional transitions to parasitism in the Harpacticoida (Fig. 2 dotted red lines). First, some species of thalestrid harpacticoids are gall-forming parasites living in the thallus of seaweeds (Ho & Hong, 1988; Shimono, Iwasaki & Kawai, 2007; Huys, 2016) and the family Thalestridae is represented here by the free-living genus Phyllothalestris. If terrestrial leaf mining and gall-forming insects are considered plant parasites, then so should these copepods that operate similarly. Second, the Canuellidae is also sampled in analyses here, but only a few taxa are parasitic in the otherwise free-living family. Of the canuellids associated with hosts, the most likely parasite is Echinosunaristes (Huys, 1995) though other genera including Intersunaristes Huys, 1995 and Sunaristes Hesse, 1867 may be parasitic. Huys (2016) considered Echinosunaristes to be non-parasitic detritivores, but we find its predilection site (the anus of sea urchins) and morphological modifications (i.e., thinner cuticle thought to be a response to its sheltered microhabitat and the reduction of oral appendages (Huys, 1995)) strongly suggestive of parasitism. Evidence for parasitism is less conspicuous in Intersunaristes and Sunaristes though their association with hermit crabs appears durable (Humes & Ho, 1969; Hamond, 1973). Third, in the Tegastidae, there are a number of copepods associated with metazoan hosts and evidence of parasitism is most compelling for Aglaogastes cnidicus (Humes, 1981; Huys, 2016) given that >1,500 copepods and 167 copepodids of A. cnidicus were collected on a single specimen of the hydroid Aglaophenia cupressina (Humes, 1981). While the Tegastidae and Peltidiidae are sister taxa in Fig. 2, both families include many free-living species which indicates these are likely separate transitions to parasitism within each family, rather than indicative of a shared parasitic ancestor.

Finally, while there are no molecular phylogenetic data for the following transitions to parasitism because they have never been sampled in a phylogenetic analysis, there is taxonomic and morphological data supporting three additional parasitic harpacticoid clades: Balaenophilus Aurivillius, 1879, the Cholidyinae (family Tisbidae), and a number of genera in the Laophontidae (Fig. 3). First, three species of Balaenophilus are found in high densities on marine tetrapods (whales, sea turtles, and manatees), and while the exact nature of their association with some of their hosts is unclear, the abundance of baleen keratin in their fecal pellets, at least, is indicative of parasitism on whales (Badillo et al., 2007). No members of Balaenophilus have been sampled in molecular phylogenetic studies, but, given that their morphology differs substantially from all the other clades of parasites and includes a highly modified clawed nauplius larva (Ogawa, Matsuzaki & Misaki, 1997), this too likely represents a unique evolution of parasitism, perhaps nested within the Miraciidae Dana, 1846 as suggested by Willen (2000). Second, of the 148 species within the mostly free-living family Tisbidae, there are 14 parasites in the subfamily Cholidyinae (Walter & Boxshall, 2021). All but one of these parasitic species encyst within deep sea cephalopods while the remaining species, Neoscutellidium yeatmani (Zwerner, 1967), parasitizes Antarctic Eelpout gills (Zwerner, 1967; Huys, 2016). Other members of the Tisbidae, particularly some species of Tisbe, may be facultative parasites given that they are commonly found in high densities on the gills of mussels, and Humes (1954) found that dislodged copepods would often return to mussel gill tissue during dissection. While none of the parasitic tisbids have been sampled in molecular phylogenetic studies, these species likely represent at least one independent transition to parasitism within the family Tisbidae. Third, there are a number of parasitic species within the Laophontidae that are conservatively counted as a single transition to parasitism here since the phylogeny of the family is poorly understood. Two species of Microchelonia Brady, 1918 are undoubtedly ectoparasites of sea cucumbers and have highly modified morphology including scraping mouthparts and clawed legs that are no longer usable for swimming ( Kim, 1991; Huys, 2016). Members of other laophontid genera are also likely parasitic. All life stages of the copepod genus Mictyricola Nicholls, 1957 are regularly found in high densities (35–82 per host) on the setae of burrowing crabs belonging to the genus Mictyris Latreille, 1806. They are poor swimmers and have not been found off their host, even in their burrows, suggesting the relationship is obligate. Even if they only scavenge on food from the host, that could fall under our definition of parasitism if it reduces the energy the host receives. A similar situation occurs in Robustunguis Fiers, 1992 on xanthid crabs, which have a highly modified, claw-like leg 1 presumably for attachment to host setae (Fiers, 1992). A number of other laophontid genera are associated with hosts such as Hemilaophonte Jakubisiak, 1933 and Paralaophonte Lang, 1944 but little is known about the nature of their relationship with their hosts. While none of the parasitic laophontids have been sampled in molecular phylogenetic studies, these cumulatively represent at least one transition to parasitism based on Microchelonia alone.

There is a large range in species richness of the 14 clades of parasitic copepods (Table 3). The transition to parasitism has resulted in a massive evolutionary radiation and in the colonization of more than 10 host phyla on two occasions: the Siphonostomatoida + Monstrilloida (2,435 species) and the peocilostome Cyclopoids (2,235 species) (Walter & Boxshall, 2021). Both of these transitions occurred >100 mya ago, though the exact timing is unclear given the limited fossils of parasitic copepods and molecular divergence time estimations. Still, trace fossils of parasitic cyclopoids in echinoids are known from at least ∼168 mya in the mid Jurassic (Radwanska & Poirot 2010) and the lone fossil siphonostomatoid is from ∼125 mya ago in the mid Cretaceous (Cressey & Paterson, 1973; Cressey & Boxshall, 1989). Both of these fossils are of rather derived members of their respective groups, which suggests the transitions to parasitism in these clades occurred much earlier. The transition to parasitism in the clade comprising the Notodelphyidae, Ascidicolidae and related families of tunicate parasites also lead to a large diversification resulting in 699 species (Walter & Boxshall, 2021). One parasitic transition encompassing the Lernaeidae + Ozmanidae appears to have occurred in freshwater and resulted in moderate species richness (129 species), perhaps as a specific response to colonizing freshwater (Ho, 1994; Walter & Boxshall, 2021). Given the diversity of freshwater fishes available to serve as hosts for the lernaeids, it is perhaps surprising there is not greater diversity in freshwater fish parasitizing copepods when compared to those parasitizing marine fishes. This could be because lernaeids are often less host specific and may have speciated less as a result, or perhaps the transition to fish parasitism in freshwater by lernaeids is recent.

While the species richness is high for some parasitic copepod clades, most transitions to parasitism in copepods have not led to large radiations in terms of species number (Table 3). In the cyclopoids, the remaining two transitions are small: there are only five known species in the Thaumotopsyllidae and Eucylops bathanalicola is the only known parasite in its genus. All parasitic harpacticoids are relatively species-poor incursions into parasitism. In the Ameiridae, there are perhaps four or more parasitic species including Nitokra bdelluare, N. spinipes and the monotypic Abscondicola and Antillesia. There are three described species of Balaenophilus (only two of which were considered valid by Huys (2016)). Within the Canuellidae, the single species of Echinosunaristes has the greatest evidence of parasitism, and perhaps the single species of Intersunaristes and the four species of Sunaristes may be parasites or commensals. There are 14 obvious parasites in the Cholidyinae. In the Laophontidae, four species are likely parasitic: two each of Mictyricola Nicholls, 1957 and Robustunguis Fiers, 1992. In the Peltidiidae, we consider the two species of Alteuthellopsis parasites plus perhaps a few other peltidiids. A number of copepods in the Tegastidae are associated with hosts, but we find available data of parasitism convincing for only A. cnidicus. In the Thalestridae, there are four species of herbivorous copepods on brown and red algae (Huys, 2016), and we consider at least the gall-inducing species parasites. Collectively, these 10 small transitions into parasitism encompass only 38 species. It is interesting to speculate if the low diversity in the harpacticoid parasitic clades is related to their more loose associations with hosts. Many harpacticoids are epibenthic predators or scavengers and these clades may represent more recent forays into parasitism.

There are no known parasitic species in the Calanoida, Gelyelloida, Misophrioida, Mormonilloida, and Platycopioida. While this is not so conspicuous in the latter 4 orders since they contain only 2–37 species each (Walter & Boxshall, 2021), the lack of parasitism in the large order Calanoida is striking (Table 2). This order contains 2,709 mostly marine species, the overwhelming majority of which are planktonic (Boxshall & Defaye, 2008; Walter & Boxshall, 2021). There are a few isolated reports of possible symbiotic relationships among the calanoids. Humes & Smith (1974) reported that Ridgewayia fosshageni Humes & Smith, 1974 formed aggregations in the vicinity of the actiniarian Bartholomea annulata (LeSueur, 1817). Although these aggregations were more stable than similar aggregations formed near rocks or near another actiniarian species, the copepods moved constantly and were never observed coming to rest either on the anemone or on the substrate. The nature of this association is uncertain. It might possibly involve the copepod feeding on mucus produced by the actiniarian since another calanoid, Acartia negligens Dana, 1849 has been demonstrated capable of feeding on mucus produced by reef corals; Richman, Loya & Slobodkin (1975) found that A. negligens can assimilate up to half of the organic matter present in the mucus. Still, these are rare exceptions: most calanoids inhabit the water column and have dual purpose mouthparts adapted for both swimming and food capture (Svetlichny & Hubareva, 2005; Svetlichny, Larsen & Kiørboe, 2020). In fact, members of the Misophrioida, Mormonilloida, and Platycopioida also exhibit mouthparts with this dual function (Sars, 1903; Boxshall, 1985). In contrast, the remaining copepod orders do not use oral appendages for swimming. Indeed, the mouthparts of parasitic copepods have lost the parts of the limb responsible for generating water flow. It may be that the locomotory function of mouthparts in calanoids (and also misophrioids, mormonilloids, and platycopioids) has constrained their morphology and made the transition to parasitism more challenging.

The 14 independent transitions to parasitism identified here are only a preliminary estimate of parasite evolution in copepods. With only 365 of the 14,485 species of copepods sampled in molecular phylogenetic analyses, most of the copepod tree remains unresolved. We suspect 14 transitions to parasitism is, if anything, an underestimate. The total number of transitions has been counted conservatively here (e.g., all poecilostomes counted as one transition, parasitic ameirids as one transition, notodelphyids and ascidicolids as one transition, etc.). While it is possible that denser phylogenetic sampling will show some of the 14 parasitic clades recovered here merge into fewer clades, it is also likely that additional independent clades of parasitism will be recovered considering that only 92 of the >5,000 parasitic copepod species have been sampled. Nonetheless, the 14 transitions to parasitism identified here are substantially more than previously published estimates.

Weinstein & Kuris (2016) recently estimated six independent transitions to parasitism in the Copepoda. The difference between our counts can be attributed to a more poorly resolved copepod phylogeny when Weinstein & Kuris (2016) made their estimate, different interpretations of parasitism versus commensalism, or the fact that Weinstein & Kuris (2016) may have been unaware of a number of the less species rich clades of parasitic copepods. We agree with five of the six transitions tallied by Weinstein & Kuris (2016): (1) Balaenophilus, (2) Cholidyinae, (3) Echinosunaristes, (4) the parasitic cyclopoids, and (5) the Siphonostomatoida and Monstrilloida. As a result of the increased resolution of the cyclopoid phylogeny by Khodami et al. (2019), it is now clear there are at least five clades of parasitic cyclopoids while Weinstein & Kuris (2016) counted only two. Strangely, one of the two they counted was the enterognathid genus Zanclopus Calman, 1908; however, phylogenetic data are lacking for this family and it has long been treated as a close relative of a cluster of other mostly-tunicate associated families (including the Ascidicolidae, Buproridae, Botryllophilidae, and Enteropsidae), so we conservatively counted this as a single transition along with the Notodelphyidae and Archinotodelphyidae. It would be interesting to sample these lineages in future phylogenetic studies to test if they represent one or more independent clades. Weinstein & Kuris (2016) discounted the Thalestridae as loosely associated and not parasitic, but we find the gall-inducing activity of some thalestrids convincing evidence of a parasitic relationship (e.g., Fahrenbach, 1962; Ho & Hong, 1988). Finally, Weinstein & Kuris (2016) were either unaware of some of the less species-rich transitions to parasitism or did not consider the following taxa parasitic: Aglaogastes cnidicus (Tegastidae), Alteuthellopsis (Peltiidae), Ameiridae, and Eucylops bathanalicola.

The notion that copepods have evolved to be parasitic at least 14 times has important implications for our understanding of the evolution of the group. There are many morphological changes often associated with transitions to parasitism in copepods, such as the reduction in body and limb setation and segmentation; dorsoventral flattening; incorporation of additional leg bearing segments into the cephalosome; cephalothorax forming a suction cup for attachment; modification of antennae and maxillipeds into robustly clawed appendages; development of root-like absorptive processes; and even the loss of all appendages and external segmentation. It is clear these have happened convergently many times. Similarly, host seeking behaviors and strategies to avoid detection by the host have also evolved multiple times, perhaps even immunomodulation. These replicate transitions to parasitism make copepods an ideal system to explore morphological evolution and genetic changes associated with the evolution of a parasitic lifestyle.

In a number of instances, clades of parasitic taxa are closely allied with taxa more loosely associated with hosts. This is true of a number of poecilostome clades, the tunicate-associated parasitic cyclopoids (i.e., Notodelphyidae + related families), and the parasitic harpacticoid clades. Even among the Siphonostomatoida, which is generally regarded as entirely parasitic, a number of taxa retain more free-living copepod morphology, and some of these appear to have looser associations with hosts, such as the hydrothermal vent associated Dirivultidae Humes and Dojiri, 1981 (see Tsurumi, De Graaf & Tunnicliffe, 2003) and Ecbathyrion Humes, 1987, as well as some asterocherids. In the context of more robust taxon sampling, it would be interesting to explore the phylogeny at the bases of these clades, and to examine host associations of these groups with ancestral state reconstruction analyses to infer potential avenues of host colonization.

What remains a major challenge to improving our understanding of parasitic copepod evolution is elucidating the nature of the association symbiotic copepods have with their invertebrate hosts, where little is known regarding the nature of their association (i.e., whether dependent or facultative, whether commensal or parasitic) (Humes, 1987; Ho, 2001; Boxshall & Halsey, 2004; Huys, 2016). Collectively there are over 2,000 species of copepods associated with invertebrate hosts of which we have almost no information on the nature of their association. It is common for the only host association data to be just that: an association (e.g., copepods collected in the tube of a polychaete; copepods collected from the washings of a stony coral). In many cases, modified morphology is a practical indicator of parasitism, such as sucking or scraping mouthparts, or robust clawed appendages for attachment, but this is far from a perfect approximation, especially for the relatively unmodified copepods associated with invertebrates. Careful observation, analysis of gut contents, isotope analysis, and perhaps metabolic or biochemical profiling are needed to inform our understanding of the relationship symbiotic copepods have with their hosts. The nature of the relationship these more loosely associated copepods have with their hosts is key to understanding the evolution of parasitism since these are possible intermediate stages in the transition from free-living to parasitic. Both a more thorough understanding of host associations and more robust taxon sampling in phylogenetic studies are needed to improve our understanding of parasitic evolution in copepods.

Conclusions

The number of transitions to parasitism, the morphological diversity, and the host range of copepods makes this group an ideal system to explore the evolution of parasitism. With 14 independent transitions to parasitism, copepods have the fourth greatest number of parasitic origins in the Metazoa, surpassed only by the Acari (31), Hexapoda (87+), and Nematoda (18) (Weinstein & Kuris, 2016). Given that 14,120 of the 14,485 copepod species have never been sampled in molecular phylogenetic analyses, this study highlights the need for additional sampling of copepod taxa, which may reveal additional origins of parasitism. The Cyclopoida, Harpacticoida, and Siphonstomatoioda require the greatest additional sampling effort (Table 2), and, coincidentally, these orders contain nearly all of the parasitic copepods. While the lack of phylogenetic information available has prevented copepods from achieving their full potential as a powerful model system for understanding parasite evolution, the evolutionary dynamics of host associations, and morphological evolution, this group is primed for the application of molecular techniques for high-level systematic revision. A robust phylogeny of the copepods will enable the revision of contentious classification of copepods, bring stability to their taxonomy, and enable the exploration of the exceptional morphological variation and host range of parasitic copepods.

Supplemental Information

Supplemental Information 1 Grafted phylogeny of only sampled taxa

Click here for additional data file.

Supplemental Information 2 Grafted phylogeny including all sampled taxa and taxonomic data from OTT for all copepod species

Click here for additional data file.

We thank all the researchers that produced the molecular phylogenies used in these analyses and we are grateful to all the copepod taxonomists that have cumulatively produced the copepod taxonomy used in the OTT, especially through the World of Copepods Database and WoRMS. We also thank Christoph Noever and two anonymous reviewers whose careful comments improved this manuscript. Any opinions, findings, conclusions, or recommendations expressed in this material are those of the author(s) and do not necessarily reflect the views of the National Science Foundation.

Additional Information and Declarations

Competing Interests

Author Contributions

Data Availability

Keith A. Crandall is an Academic Editor for PeerJ.

James P. Bernot conceived and designed the experiments, performed the experiments, analyzed the data, prepared figures and/or tables, authored or reviewed drafts of the paper, and approved the final draft.

Geoffrey A. Boxshall and Keith A Crandall conceived and designed the experiments, analyzed the data, authored or reviewed drafts of the paper, and approved the final draft.

The following information was supplied regarding data availability:

All phylogenies are available in the Open Tree of Life curatorial system: https://tree.opentreeoflife.org/curator/profile/jbernot/copepoda.

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
