# Peer review of "A synthesis tree of the Copepoda: integrating phylogenetic and taxonomic data reveals multiple origins of parasitism"

_PeerJ, doi:10.7717/peerj.12034_

## Round 0.1 · original submission · Major Revisions

I agree with reviewers that overall, this manuscript is well-written and gives valuable insights into the potential origins of parasitism in copepods. However, in addition to the reviewers' comments, I would suggest the authors improve on the Methodology section. In particular, kindly provide details on what were the selection criteria during literature trimming (from 178 to 26) and what data was extracted from the selected published literature.

Reviewer 1 ·

Basic reporting

no comment

Experimental design

no comment

Validity of the findings

no comment

Additional comments

I enjoyed reading your manuscript since it is well written and matches my research interests. My comments are mostly minor in nature, and I am happy to recommend publication and revision.

Line 20 you wrote 14,397 species, but in line 40 “14,397 valid species”, please check.

Many citations are missing through the manuscript. For example, lines 48 and 53 in the Introduction.

Line 57, Please note there are some overlaps between these groups.

Line 75, If I understood correctly, these phylogenies/analyses were not for copepods. I would suggest to make a smooth transition with last paragraphs.

Line 94, why is interesting/important to study parasitic copepods? You need to introduce it.

Line 93-100, I would move them to the Method (how you identify the parasitic copepods).

Line 105, are you sure you did it on June 01, 2021?

Line 106 and others, I think you need citations for the software.

I would suggest to move the Figure 1 into supplementary material, as it is not your results. Otherwise, I would suggest you provide a paragraph to describe how you did the morphological analyses.

Line 465-466, why did the “transitions to parasitism” have important implication to understand the morphological evolution? Please clarify.

Line 476, as you did not touch anything about the morphology of copepods in this study, I would suggest you to be very carefully to discuss this issue.

I would suggest you to largely reduce the Discussion (the current version is too long): you can focus the “transitions to parasitism”, after a short discussion of “copepod phylogenies”.

Reviewer 2 ·

Basic reporting

The manuscript under revision is a nice piece of work dealing with evolution of parasitism in copepods. The work is based on published phylogenetic analyses merged to produce a synthetic phylogeny of copepods. On such a phylogeny, the possible transition/s to parasitism have been mapped.
The manuscript is very well written and it was a pleasure to read it, nonetheless, I am under the impression that the work is too preliminary. As the Authors state several times through the text, there is a non-negligible lack of knowledge for a conspicuous number of copepod species. The under-representation in terms of species within families and orders is the main drawback I see. The second issue I recognise in the work is a lack of statistical treatment of some results such as for instance the possible number of transitions to parasitism in copepods. PAUP was used to carry out parsimony analyses, but both analyses gave equivalent results. The Authors sublimely discussed this result, but indeed, to me, that is not a conclusive result. I would suggest to run other tests using other approaches (Bayesian statistics?) in order to validate one or the other hypothesis.
I feel that this work is too preliminary and it leaves more open questions than it addresses. What I learned from the work presented in the manuscript under revision is that our knowledge about copepod evolution and the evolution of parasitism is too scanty. The present manuscript looks more like a ‘position paper’ or the preliminary results of a grant application than a regular research paper. The Authors state (lines 531-533) ‘The Cyclopoida, Harpacticoida, and Siphonstomatoioda require the greatest additional sampling effort (Table 2), and, coincidentally, these orders contain nearly all of the parasitic copepods.’ This made me think that the bulk of data in molecular phylogeny is too weak to carry out a work like that presented in the manuscript. In the manuscript, contrasting hypotheses are validated and rejected in the same sentence, which is good and reveals a strong thinking effort the Authors have carried out to draw conclusions. Nevertheless, this weakens the conclusions in my opinion.

Experimental design

The research question is well defined and an investigation on transition to parasitism in copepods is needed. Nevertheless, as stated above, I think that the knowledge available to date to carry out the analyses in the present manuscript is too scanty.

Validity of the findings

See the comments in section 1

Additional comments

Lines 49-51. DNA extraction from formalin preserved samples is rather tricky, although a paper was published 17 years ago describing a protocol to extract DNA from formalin samples. The yields are somehow low (70% of gDNAs produced a PCR amplification on 91bp fragments and 40% on 181bp fragment; n=20; https://doi.org/10.1016/j.ympev.2003.11.002). For other organisms, DNA extraction from formalin preserved samples is almost a routine (DOI: 10.2144/97223bm03; https://doi.org/10.1186/s13104-016-2140-1; etc.) I think that this is worth a mention otherwise the reader would have partial information about that.
Lines 52-54. References are needed here.
Lines 55-56. I do not understand what the Authors mean by this sentence: ‘Overall, there are few accurately identified reference sequences of copepod taxa.’
Lines 59-60. I do not understand the correlation between the five ecological groups and the ‘[…] limited the integration of copepod phylogenies into a cohesive tree of life.’
Line 106. Mosquite was updated to V3.61 in December. Which version did you (3.31 if I am right) use and why? Mosquite website suggests several citations for the software, please have a look at that and cite Mosquite as requested (https://www.mesquiteproject.org/How%20to%20Cite%20Mesquite.html?PublishingResultsPanel=open)
Lines 135, 196, 236. Figure 3 does not contain panels, so I do not understand what do Fig.3A and B mean.
Lines 141-156. I do not understand how the Authors chose the published studies to be included in their synthetic phylogeny analyses. There are some recent papers that have not been included although morphology and multi-locus BI and ML phylogenies were carried out, especially within Cyclopoida. Five Oncaea and one Triconia species were relatively recently investigated by COI and ITS1-5.8S-ITS2 rDNA (https://doi.org/10.1371/journal.pone.0175662). Even more interestingly, Cyclopoida contain the debated suborder formerly known as Peocilostomatoida and I would be curious to see where Oncaea would sit in your synthetic phylogeny. If it would cluster together with Poecilostomatoida then some more debate can be done on the monophyly of Poecilostomatoida.
Line 242. There are no black branches in Fig. 3 (A ???)
Lines 285-295. I think that if parsimony is not completely resolving, other statistical methods may be used (?). This part is the real core of the work and upon this result conclusions are drawn. I would try to make this part more robust.
Lines 298-301. The disclaimer about the number of molecular markers and number of studies on

I would suggest to turn the manuscript into a position paper and to dim conclusions on the evolution of parasitism. I do not know whether the Authors intend to follow-up this topic in their laboratories, but reading the manuscript it looks like.

·

Basic reporting

no comment

Experimental design

no comment

Validity of the findings

no comment

Additional comments

The article gives valuable insight and significantly improves our understanding of the evolution of parasitism within copepods. The article is well composed and gives a solid overview of the current knowledge on evolutionary pathways into parasitism in the taxon.


Below are detailed comments that should be addressed:

Line 51 - 53: A limited assortment of markers and the use of a selected gene set is not unique for studies on copepods. As important as the number of markers is the selection of the right markers based on how conserved they are and what phylogenetic level is investigated.

Line 103 - 105: It should be added how the final selection of 26 publication was done from the 178 search results from Google Scholar.

Line 114, 116, 119: File names are given, but they do not seem to be linked to accessible files. Those names (e.g. labelled_supertree_simplified_ottnames.tre) should not be included in the text if not published as supplementary data.

Line 132: Repetition from Materials and Methods.

Line 135: “(Fig 3A, B)” There is no separation into A/B in this figure.

Line 148/149: “The species-poor orders Gelyelloida, Misophrioida, and Monstrilloida, have been relatively well-sampled.”
According to Table 2, no Gelyelloida or Misophrioida are included in the synthesis tree, and while constituting 1.7% of species diversity in Monstrilloida, only a low number of 3 monstrilloid taxa are included in the synthesis tree.

Line 207: The word “although” is not fitting here. I suggest to split the sentence up into two. E.g.: “Another species rich group are the…”.

Section line 285 - 306: The authors consider the non-parasitic taxa within the parasitic poecilostome cyclopoids unlikely to be derived from parasitic forms, following the hypothesis of parasitism being an irreversible transition, related to Dollo’s law of irreversible evolution. However, more recent studies than cited found exceptions to this rule, and copepods might include further exceptions. This scenario should in particular be considered since parasitic copepods in this group do not display the most extreme morphological modifications due to parasitism compared to other parasitic copepod taxa such as for example those shown in figure 1.
e.g.:
Klimov, P.B. and OConnor, B., 2013. Is permanent parasitism reversible?—Critical evidence from early evolution of house dust mites. Systematic biology, 62(3), pp.411-423.
Weinstein, S.B. and Kuris, A.M., 2016. Independent origins of parasitism in Animalia. Biology Letters, 12(7), p.20160324.

Line 323 - 324: It is unclear what “80 mm land crabs” refers to. Presumably this is carapace width, but this should either be explained or just left out since not really relevant here. “Found with high infection rate at 100% prevalence” might be sufficient.

Line 372: Missing “of” between “none” and “the”.

Line 466 - 472: From this sentence it appears that all the exemplified morphological changes, including the most extreme examples such as loss of all appendages, have happened convergently in all 14 transitions. This sentence has to be rewritten in order to clarify that not all modifications appear in each parasite clade.

As a more general comment; it is striking that that Calanoida do not contain any parasitic forms, despite being one of the most speciose copepod orders. It could be interesting to discuss why parasitism evolved multiple times in all other major copepod clades but not in calanoids, such as ecological or morphological preadaptations to parasitism. However, this is not crucial to the quality of the paper.

Table 3: Species numbers for Siphonostomatoida and Monstrilloida should be given individually instead of combined. In addition, the combined species numbers for these two orders slightly deviate from the numbers in table 2.

---

## Round 0.2 · accepted · Accept

The revision done by the authors was thorough and the current version of the manuscript is suitable for publication.

Reviewer 1 ·

Basic reporting

I am happy with the revisions. Thank you.

Experimental design

no comment

Validity of the findings

no comment

Additional comments

no comment

Reviewer 2 ·

Basic reporting

The authors have clearly addressed all my points and, although I still find some criticisms to the procedure (e.g. the criteria for exlusion or inclusion based on e.g. concatenation https://doi.org/10.1080/10635150601146041), I am happy to support publication. I would have appreciated a more thorough investigation on the ancestral state but, because one of the aims of the work is to quantify the lack of knowledge in order to stimulate more phylogenetic studies in copepods (the Authors state) then I am happy to promote its publication.

Experimental design

.

Validity of the findings

.

Additional comments

.

·

Basic reporting

no comment

Experimental design

no comment

Validity of the findings

no comment

Additional comments

All comments from the first review have been well addressed.